# A Simple Pre-Operative Nuclear Classification Score (SPONCS) for Grading Cataract Hardness in Clinical Studies

**DOI:** 10.3390/jcm9113503

**Published:** 2020-10-29

**Authors:** Jorge Mandelblum, Naomi Fischer, Asaf Achiron, Mordechai Goldberg, Raimo Tuuminen, Eran Zunz, Oriel Spierer

**Affiliations:** 1Department of Ophthalmology, Tel Aviv Sourasky Medical Center, Tel Aviv 6243906, Israel; jorgemandelblum@msn.com (J.M.); naomi797@hotmail.com (N.F.); eranzunz@gmail.com (E.Z.); 2Sackler Faculty of Medicine, Tel Aviv University, Tel Aviv 6997801, Israel; achironasaf@gmail.com; 3Bristol Eye Hospital, University Hospitals Bristol NHS Foundation Trust, Bristol BS1 2LX, UK; 4Mid and South Essex NHS Foundation Trust, Basildon SS16 5NL, UK; motalegol@gmail.com; 5Helsinki Retina Research Group, University of Helsinki, 00100 Helsinki, Finland; raimo.tuuminen@helsinki.fi; 6Department of Ophthalmology, Kymenlaakso Central Hospital, 48210 Kotka, Finland; 7Department of Ophthalmology, Wolfson Medical Center, Holon 5822012, Israel

**Keywords:** cataract, grading, classification, nuclear color, phacoemulsification

## Abstract

Background: The aim of this study was to evaluate whether a simplified pre-operative nuclear classification score (SPONCS) was valid, both for clinical trials and real-world settings. Methods: Cataract classification was based on posterior nuclear color: 0 (clear), 1 (subcapsular/posterior cataract with clear nucleus), 2 (mild “green nucleus” with plus sign for yellow reflection of the posterior cortex), 3 (medium “yellow nucleus” with plus sign for brown/red posterior cortex reflection), 4 (advanced with 4 being “red/brown nucleus” and 4+ white nucleus), and 5 (hypermature/Morgagnian nucleus). Inter- and intra-observer validity was assessed by 30 Ophthalmologists for 15 cataract cases. The reliability of the cataract grading score in a surgical setting was evaluated. Correlation of nuclear scores was compared with phacoemulsification cumulative dissipated energy (CDE) in 596 patients. Results: Analysis of mean intra-observer Cohen kappa agreement was 0.55 with an inter-observer score of 0.54 for the first assessment and 0.49 for the repeat assessment one week later. When evaluating results by nuclear color alone, there was a substantial agreement for both the intra-observer (0.70) and inter-observer parameters: 0.70 for the first test, and 0.66 on repetition with randomization of the cases after a week. CDE levels were found to be significantly different between all SPONCS score groups (*p* < 0.001), with a lower CDE related to a lower SPONCS score. A strong correlation was found between the SPONCS score and CDE (Spearman′s rho = 0.8, *p* < 0.001). Conclusion: This method of grading cataract hardness is both simple and repeatable. This system can be easily incorporated in randomized controlled trials to lower bias and confounding effects regarding nuclear density along with application in the clinical setting.

## 1. Introduction

Cataract surgery is one of the most common procedures performed in the United States [1]. A robust cataract “lens hardness” classification is necessary in these trials in order to reduce bias and confounding factors. Several lens hardness classifications have been proposed for this purpose [2,3,4,5,6,7,8,9]. Variations of these classifications are widely used in clinical practice and research. However, the accepted, standardized classifications tend to be complex and time-consuming. Most current grading systems are detailed classifications which are based on standardized images [2,3,4,5,6,7,8,9] (Table 1). Indeed, these grading systems are mainly used in research and seldom used in clinical practice. The most widely used system is the Lens Opacities Classification System (LOCS) III, a chart consisting of six slit-lamp images of nuclear color and opalescence, five cortical retro-illumination images, and five retro-illumination images of posterior subcapsular cataracts [2]. Other grading systems, such as the Oxford system, also employ analysis of a large number of cataract characteristics, making it difficult to apply in clinical practice [5]. The classification system proposed by the Japanese Cooperative Cataract Epidemiology Study Group is based on clinical photos of cortical opacities, nuclear opacity, and subcapsular opacities [6]. The World Health Organization’s (WHO) simplified classification is solely based on nuclear opalescence [7]. However, unlike our proposed system, it is still based on a comparison to standardized photographs. The recent BCN 10 grading system for nuclear cataracts is based on nuclear opacity, which is similar to our proposed classification. A clear lens is classified as NO, with cataract grading scores from N1 to N10 [8]. Nevertheless, this grading system, based on reference photograph color images, categorized 10 different levels of nuclear opacity, making it clinically ungainly to use. Recently, there have been advances in imaging technology and deep learning, which have helped in developing reproducible cataract grading [9,10,11,12].

Cataract severity (“lens hardness”) is an important consideration in the pre-operative surgical plan and the operative phacoemulsification indices of femtosecond-assisted procedures. Optimal phacoemulsification energy and prudent use of viscoelastic materials reduces the risk of substantial corneal endothelial loss and detrimental visual outcomes [13].

A clinician needs a quick, easy grading system which will help to guide surgical decisions and parameters. Research requires a lens grading system which is accurate and reproducible. An accurate, simple grading system for research purposes which can be applied to the clinical setting is more useful and accessible for guiding “real-world” practice. We developed a cataract classification which is based on nuclear color alone and is not dependent on standard photos. The aim of this study is to investigate the validity of this simplified grading scale, both as a classification tool and regarding its correlation to phacoemulsification energy.

## 2. Experimental Section

### 2.1. Materials and Methods

The study was approved by the Institutional Review Board in compliance with the Declaration of Helsinki (protocol code number: 0735-15, version 2:0, approval date: 7 March 2016).

### 2.2. Simple Pre-Operative Nuclear Classification Score (SPONCS)

The Standard Classification was conceptualized by the first author (JM). It was based on posterior nuclear color on a scale of 0–5. A plus sign was added if the posterior cortex color showed findings of a more advanced cataract. These are described in Table 2.

Photographic images of the lens (using the Canon EOS slit lamp photography at 45 degrees) were evaluated (Figure 1).

The photographs were obtained from a selection of pre-operative images taken by the author (JM). All identifying patient details were removed. The pictures were selected by the primary co-authors (JM and NF). All chosen pictures had agreement regarding their grading. They were chosen based on image quality. In order to minimize bias, the first images found to meet this standard were those selected to be included in the questionnaire.

Ophthalmologists at the institution with at least one year of training were introduced to the new cataract grading tool. They were then given a set of 15 photographs of cataracts to grade (Time 1). The photographs were viewed projected onto a screen by an Epson WXGA projector using a Dell Latitude 5480 computer. An example photograph of each cataract grade was given prior to the questionnaire. The questionnaire consisted of two photographs of each grade (except for grade 5 where there was only one image available). These were displayed in random order. After a period of 1 week, they were retested (with the same photographs in a different order) to test for intra-observer variation (Time 2).

### 2.3. Validation Study

In order to evaluate the SPONCS classification system and to validate the score, we conducted a cohort study of patients undergoing routine cataract phacoemulsification surgery. In total, 596 eyes of 596 patients were included in the study. Surgeries were performed by a single surgeon (JM), with the Alcon Infinity phacoemulsification machine. The patients were a cohort from the years 2011–2016 who were graded and examined pre-operatively by the surgeon (JM). From this cohort, 599 patients were evaluated at random. Three patients were excluded due to missing data. Phacoemulsification cumulative dissipated energy (CDE), a phacoemulsification unit parameter designed to monitor the amount of energy delivered during phacoemulsification, was recorded for each surgery.

### 2.4. Statistical Analysis

Statistical analysis was performed using SPSS (IBM corp. Released 2017. IBM SPSS Statistics for Windows, Version 25.0. Armonk, NY: IBM Corp., USA). Inter-observer reliability was evaluated using Cohen’s kappa coefficient for rater agreement [14]. For the purpose of analysis, the + signs were converted to increments of 0.5 (i.e., 2+ = 2.5, 3+ = 3.5, 4+ = 4.5). Sub-analysis was also performed based on nuclear color alone. For the validation study, data are presented as mean ± standard deviation (SD). Spearman′s rank correlation (rho test) was performed to correlate the CDE and SPONCS scores.

## 3. Results

### 3.1. Inter-Observer Reliably

Thirty ophthalmologists participated in the study. The distribution of their grading at Times 1 and 2 for each picture is shown (Figure 2, Figure 3, Figure 4 and Figure 5). Analysis of intra-observer pair-wise comparisons showed a mean Cohen kappa co-efficient of 0.55 ± 0.19 (inter-quartile range: 0.40–0.67). Analysis of agreement by grouping of nuclear color only (disregarding the posterior cortex reflex) showed substantial agreement: mean 0.70 ± 0.18 (inter-quartile range: 0.57–0.85). Analysis of inter-observer variability of the graded pictures at each time point showed a Kappa coefficient of 0.54 for Time 1 and 0.49 for Time 2. There was a substantial agreement when analyzing nuclear color only of 0.70 for Time 1 and 0.66 for Time 2.

### 3.2. SPONCS Validation

A total of 596 cases were included in this study. Cohort baseline characteristics are presented below (Table 3). The mean age of the cohort was 74.2 ± 10.1 years, and the male to female ratio was 1.31:1. The majority of patients had a SPONCS score of 2 or 3 (483 patients (81% of total cases)). The cataract procedure was performed as a stand-alone procedure with a routine phacoemulsification technique. The mean CDE was 8.2 ± 6.2. CDE levels were found to be significantly different between all SPONCS score groups (*p* < 0.001), with a lower CDE related to a lower SPONCS score. A strong correlation was found between the SPONCS score and CDE (Spearman’s rho = 0.8, *p* < 0.001), as presented in Figure 6.

## 4. Discussion

In this study, we evaluated the validity of the “Standard Pre-Operative Nuclear Classification System (SPONCS)”. We found that including details of both nuclear color with the color of the posterior cortical reflex yielded only a moderate agreement rate. However, cataract grading by nuclear color alone (without the posterior cortical reflex color) increased both the inter- and intra-observer agreement to a substantial one [14]. CDE levels correlated with SPONCS scores, which verifies its usefulness in surgical planning and clinical decisions.

Cataract classification by nuclear color may be a simpler grading system, based on slit-lamp examination rather than standardized photography. It is quicker to perform than the more complex systems and does not require use of ancillary charts or diagrams.

It was impractical to bring in 15 patients representing all the SPONCS cataract grades to be examined by 30 surgeons simultaneously. Photographs were used so that all the cataract grades could be evaluated for the purpose of the study. The validity study was graded by slit-lamp evaluation. It supports how this grading is credible when looking at both photographs and slit-lamp evaluation. However, this may have caused some additional variations, and is a limitation of the study. Another limitation is that the observer was not asked about their perception of the grading system, which would have added more insight into its use. The clinical importance of nuclear opacity in femtosecond laser-assisted cataract surgery was investigated using the Pentacam Nucleus Staging (PNS) lens densitometry program to analyze cataract density and grading for Centurion phacoemulsification [15]. It showed a correlation between intra-operative phacodynamics and cataract level grading. Garcin et al. recently published their correlation of Optical Quality Analysis system (OQAS Visiometrics) parameters with surgical parameters for age-related cataracts [16]. The OQAS provides objective measurements of the image formed onto the retina. It combines quantification of ocular media transparency and optical aberrations [16]. They found that OQAS parameters significantly correlated with CDE and ultrasound time, only for pure nuclear cataracts. This supports how in a surgical setting, the most important consideration is the nuclear hardness. Davison and Chylak investigated the use of LOCSIII and phacoemulsification [4]. They found that the phacoemulsification time correlated well with nuclear, but not cortical or posterior subcapsular cataracts. An exponential increase in phacoemulsification energy was used intraoperatively as nuclear grades increased.

Our proposed grading system is a simplified classification with no need for reference photographs or additional technology. It was shown to be valid in the surgical setting, with correlating CDE scores. The best validity was seen when grouping the cataracts by nuclear color alone without the cortical component. This can be easily remembered as it follows reverse “traffic light” colors, that is, mild = “green” cataracts, moderate = “yellow” cataracts, and advanced = “red/brown”, with clear and black or Morgagnian cataracts being in separate groupings. This straightforward model should be invaluable in deciding cataract operation parameters without the need for standardized photographs or expensive lens densitometry programs. This cataract grading requires further studies in means of validation and correlation with surgery phacoemulsification parameters.

## 5. Conclusions

The SPONCS cataract grading system is a simple and efficient way of grading cataract hardness. Inter- and intra-observer agreement showed substantial agreement of this classification. It can be remembered by the “reverse traffic light” colors without the need for standard photographs, leading to greater clinical applications.

## Figures and Tables

**Figure 1 jcm-09-03503-f001:**
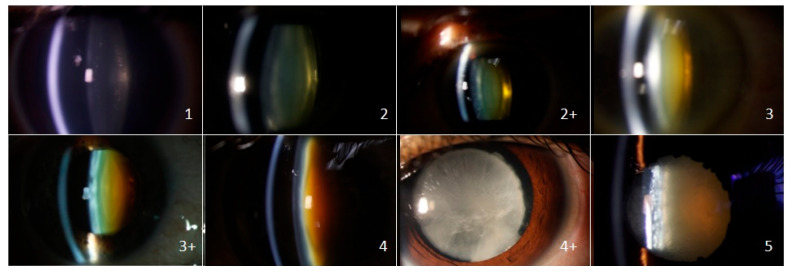
Standard Pre-Operative Nuclear Classification System (SPONCS) Canon EOS slit-lamp photographic examples.

**Figure 2 jcm-09-03503-f002:**
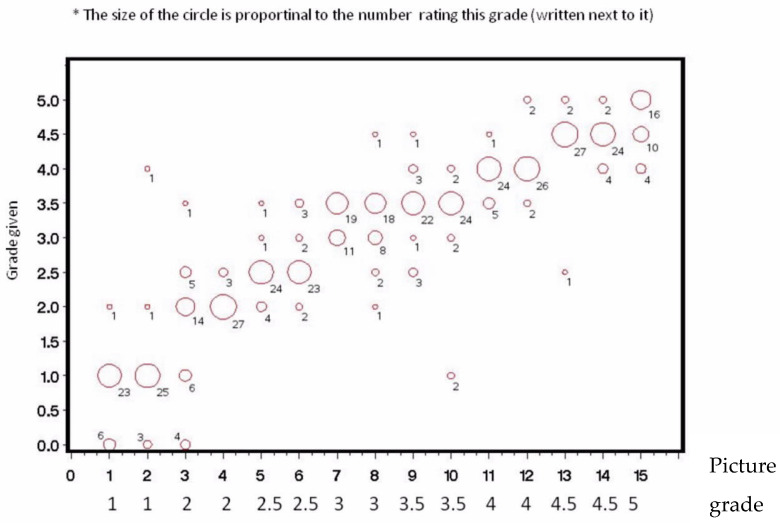
Distribution of 30 ophthalmologists’ grading for the 15 photographs at Time 1.

**Figure 3 jcm-09-03503-f003:**
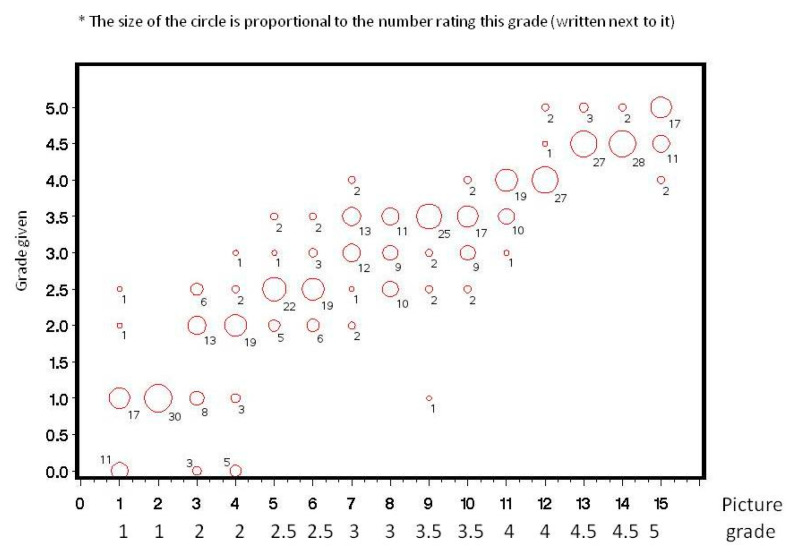
Distribution of 30 ophthalmologists’ grading for the 15 photographs at Time 2.

**Figure 4 jcm-09-03503-f004:**
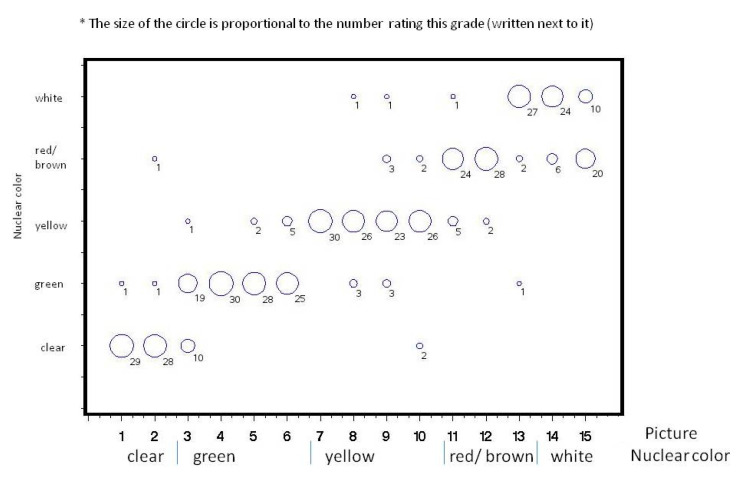
Distribution of 30 ophthalmologists’ grading when grouped according to nuclear color for the 15 photographs at Time 1.

**Figure 5 jcm-09-03503-f005:**
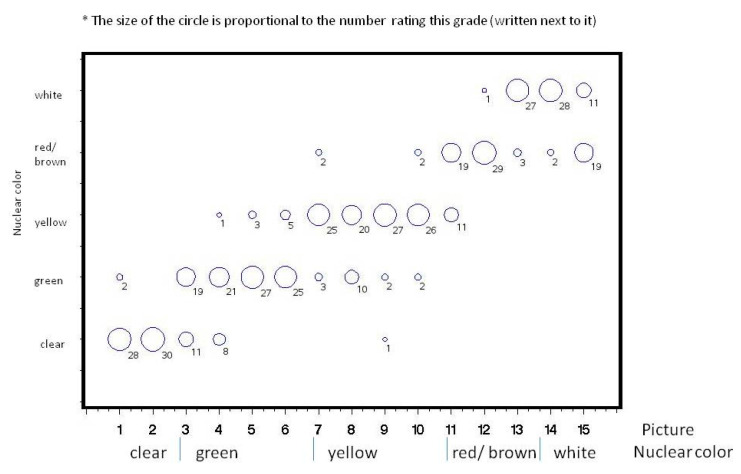
Distribution of 30 ophthalmologists grading when grouped according to nuclear color for the 15 photographs at Time 2.

**Figure 6 jcm-09-03503-f006:**
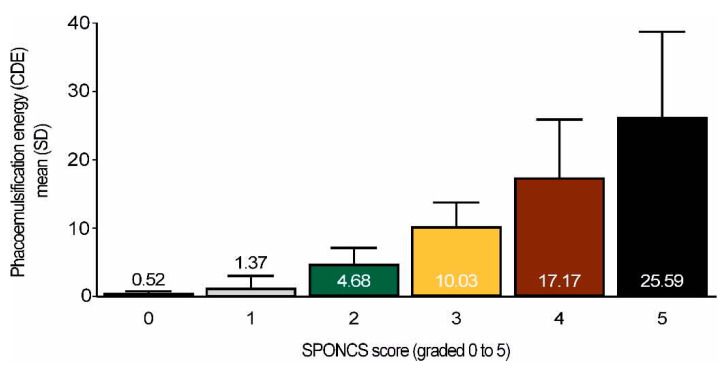
Phacoemulsification energy (CDE) levels according to the different Standard Pre-Operative Nuclear Classification System (SPONCS) scores. Mean CDE levels are presented in the bars. Spearman′s rho = 0.8, *p* < 0.001.

**Table 1 jcm-09-03503-t001:** Current cataract classification system.

Classification System	Classification Method	Year	Advantages	Limitations
Oxford Clinical Cataract Classification and Grading System [5]	Composite Slit-Lamp-Based System. Cataract Features Are Classified Morphologically, and Individual Features Are Graded by Comparison with Standard Diagrams Mounted Adjacent to The Slit-Lamp.	1986	Very Detailed	Requires a Large Number of Cataract Characteristics. Complex.
Japanese Cooperative Cataract Epidemiology Study Group [6]	Clinical Photos of Nuclear, Cortical, and Subcapsular Opacities	1990	Based on Standardized Images	Designed for Epidemiological Studies. Need to Use Standardized Photograph Reference and Analyze Multiple Lens Characteristics
Lens Opacities Classification System (LOCS) III [2]	Six Slit-Lamp Images of Nuclear Color and Opalescence, Five Retro-Illumination Images of Cortical, and Five Retro-Illumination Images of Posterior Subcapsular Cataract	Current Gold Standard 1993	Comprehensive and Detailed. Simplified in Comparison with Previous Classifications	Requires Reference Photographs. Difficult to Apply in Clinical Settings
World Health Organization (WHO) Simplified Cataract Classification [7]	Comparison to Standardized Photographs	2002	Separate Grading for Nuclear, Cortical, and Posterior Subcapsular Cataracts	Designed for Epidemiological Studies. Need for Reference Photographs
BCN 10 [8]	Reference Photograph Color Images	2017	Designed to Predict Lens Hardness Before Surgery. Ten Grades of Nuclear Opacity	Need for Reference Photographs
Artificial Intelligence (AI) [9,10,11,12]	Imaging Technology and Deep Learning		Based on Automated Optical Imaging Devices	Need for High Technology Measures. Many Algorithms. No Current Gold Standard

**Table 2 jcm-09-03503-t002:** Description of the Simple Pre-operative Nuclear Classification System (SPONCS).

Grade	Description	Nucleus Color	Posterior Cortex Color
0	Clear Lens	Clear	Clear
1	Subcapsular Cataract with Clear Nucleus	Clear	Clear
2	Mild Hardness	Green	Green
2+		Green	Yellow
3	Moderate Hardness	Yellow	Yellow
3+		Yellow	Red/Brown
4	Advanced Hardness	Red/Brown	Red/Brown
4+		Red/Brown	White
5	Hypermature/Morgagnian (Liquefaction of the Cortex and Sinking of The Nucleus to the Bottom of the Capsular Bag)	Black/White	Black/White

**Table 3 jcm-09-03503-t003:** Cohort baseline characteristics.

Variable	Category	Summary
Total, n (%)		596 (100)
Age (years)	Mean ± SD	74.2 ± 10.1
	Median (IQR)	76 (68–81)
Gender, n (%)		
	Male	258 (43.3)
	Female	338 (56.7)
SPONCS ^○^ Score, n (%)		
	0	2 (3)
	1	42 (7)
	2	247 (41.4)
	3	236 (39.6)
	4	62 (10.4)
	5	7 (1.2)

^○^ Standard Pre-Operative Nuclear Classification System (SPONCS).

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
