# Peer review of "A Simple Pre-Operative Nuclear Classification Score (SPONCS) for Grading Cataract Hardness in Clinical Studies"

_jcm, 2020, doi:10.3390/jcm9113503_

Round 1

Reviewer 1 Report

Thank you very much for the interesting paper, I added my comments and suggestions to the pdf document

Author Response

Reviewer #1 Point 1:

The introduction is very short and vague. It would be good to provide a short overview about grading and different methodologies (clinical, photographic etc.) to inform the uninitiated reader

Response: We have edited the introduction, according also to comments #3, #15, #19, #20 and #21.

Reviewer #1 Point 2:

L40-43: These are general, trite statements and do not provide substantial information about the study

Response: We have removed these lines

Reviewer #1 Point 3:

It would be good to distinguish between research and clinical needs in more detail. What does a researcher require from a lens grading system, and what is needed by clinician?

Response: A clinician needs a quick, easy grading system which will help to guide surgical decisions and parameters. Research requires a lens grading system which is accurate and reproducible. An accurate simple grading system for research purposes which can be applied to the clinical setting is more useful and accessible for guiding "real-world" practice. We have clarified it in the Introduction (lines 71-74).

Reviewer #1 Point 4:

Some pertinent information is lacking (Experimental Section)

Response: Experimental Section has been added in the Methods.

Reviewer #1 Point 5:

Could you please provide information about the SPONCS, who was involved in the classification system? Also, presentation in a table of the different grades + lens photos would be easier to get an overview, instead of presenting in running text.

Response: Thank you for these suggestions. The Standard Classification was conceptualized by the first author (JM). Accordingly, the data is now presented in table and lens photos (lines 88-101, updated Figure 1 and additional Table 2).

Reviewer #1 Point 6:

What is meant with "the lens"? Of selected patients from eye departments? From web-based resources? Who took the pictures?

Response: These photographs came from a selection of pre-operative images taken by  the author (JM). All identifying patient details were removed. This has been added to the text (lines 103-104).

Reviewer #1 Point 7:

Who selected and decided about the grade of the pictures? Could you provide details in how far bias was avoided?

Response: The pictures were selected by the co-authors (JM and NF). All chosen pictures had agreement regarding their grading. They were chosen based on image quality. In order to minimize bias, the first images found to meet this standard were those selected to be included in the questionnaire (lines 106-107).

Reviewer #1 Point 8:

The grading is missing in these 2 pictures

Response: This has been corrected for these pictures (Figure 1).

Reviewer #1 Point 9:

It is understood that you converted + to 0.5 for statistical analysis, but following the description of the grading score with + would be less confusing

Response: This has now been amended (Figure 1).

Reviewer #1 Point 10:

In how far was this a cohort study?

Response: The patients were a cohort from 2011-2016 who were graded and examined by the surgeon (JM). 599 patients from this cohort were evaluated at random. Three patients were excluded due to missing data (lines 120-122).

Reviewer #1 Point 11:

Who examined and graded the patient? The surgeon?

Response: The surgeon (line 121)

Reviewer #1 Point 12:

I cannot find any results of the many statistical tests that you alluded to?

Response: Thank you for noticing this. The Statistics in the Methods was revised to only encompass the tests that were used (lines 128-131).

Reviewer #1 Point 13:

Who scored the patients?

Response: The surgeon (line 121)

Reviewer #1 Point 14:

please reorganize the figure, it is difficult to follow

Response: Figure 6 has been re-formatted

Reviewer #1 Point 15:

Some of the information provided here should be moved to the introduction. More detailed discussion would better inform about the relevance of your results

Response: Thank you for this comment. The relevant information guided by your other comments (comment #1, #19, #20, #21) have been moved to the Introduction. The Discussion has been expanded. 

Reviewer #1 Point 16:

Which results exactly support your hypothesis that the grading system is "simple". Strictly spoken you tested observer agreement, and agreement could be theoretically also high for complex tools (although less likely). Did you ask the observer about their perception of the grading system

Response: We have not asked the observer about their perception of the grading system. We do thing that this grading system is quicker to perform than the more complex systems mainly mainly because it does not require using ancxillary charts or diagrams. This has now been addressed (lines 178-188). We also changed from "simple" to "may be simpler.

Reviewer #1 Point 17:

I do not understand why it was not practical to perform slit-lamp examinations?

Response: Thank you for this comment. It was impractical to bring in 15 patients representing all the SPONCS cataract grades to be examined by 30 surgeons simultaneously. Photographs were used so that all the cataract grades could be evaluated for the purpose of the study. The validity study was graded by slit-lamp evaluation. It supports that this grading is credible when looking at both photographs and for slit-lamp evaluation Following the reviewer comment we have clarified this (lines 181-188).

Reviewer #1 Point 18:

The LOCS III and its components of nuclear, cortical and subcapsular opacities might be more complex, but these are the basic elements of a lens examination. It would be good to discuss more why you feel that focusing on the nucleus alone would provide sufficient information instead of training ophthalmologists in comprehensive lens examination. I could understand if you propose your system for example for ancillary eye health personal in low resource settings but not for fully trained ophthalmologists?

Response: Nuclear hardness is considered the most important parameter in the surgical setting. Pertinent research has been added to the discussion (lines 215-224).

Reviewer #1 Point 19:

An overview about the pros and cons of existing tools should be presented in the introduction, maybe as a table

Response: Following this remark we now compare in a table between the existing grading tools (Table 1).

Reviewer #1 Point 20:

This information should be also moved to the introduction

Response: Thank you. This information has now been moved to the Introduction.

Reviewer #1 Point 21:

Appendix a. A short overview of the information here would inform the introduction.

Response: Thank you for this comment. We have restructured and moved this information to the Introduction.

Reviewer 2 Report

The article is interesting, very well written, and understandable. It gives information about the possibility of grading cataract hardness in both a simple and repeatable manner. The presented system can be easily incorporated in randomized controlled trials to lower bias and confounding effects regarding nuclear density along with application in the clinical setting.

All information is very accurate and well explained. 

The presented results are clear, very well presented, and validated,

The only minor revision that I suggest is to add more references regarding the subject of this paper.

In conclusion, this article could suggest a useful tool for clinical studies application of this evaluation system.

Author Response

Thank you for you insightful comments and the opportunity to submit a revised version.

Point 1:The only minor revision that I suggest is to add more references regarding the subject of this paper.

Response: Thank you for this comment. We have revisited the current literature and added relevant  references (references 17, 18).